# Design and Research of All-Terrain Wheel-Legged Robot

**DOI:** 10.3390/s21165367

**Published:** 2021-08-09

**Authors:** Jianwei Zhao, Tao Han, Shouzhong Wang, Chengxiang Liu, Jianhua Fang, Shengyi Liu

**Affiliations:** 1School of Mechanical Electronic & Information Engineering, China University of Mining and Technology-Beijing, Beijing 100089, China; ZQT1900401010G@student.cumtb.edu.cn (T.H.); ZQT1900401022G@student.cumtb.edu.cn (C.L.); SQT1900401004@student.cumtb.edu.cn (J.F.); SQT2000402039@student.cumtb.edu.cn (S.L.); 2Beijing Special Engineering and Design Institute, Beijing 100028, China; 18901181265@189.cn

**Keywords:** wheel-legged robot, complex terrain, kinematic mode, obstacle-crossing analysis

## Abstract

Aiming at the crossing problem of complex terrain, to further improve the ability of obstacles crossing, this paper designs and develops an all-terrain wheel-legged hybrid robot (WLHR) with strong adaptability to the environment. According to the operation requirements in different road conditions, the robot adopts a wheel and leg compound structure, which can realize the transformation of wheel movement and leg movement to adjust its motion state. The straight and turning process of the robot is analyzed theoretically, the kinematics model is established and solved, and obstacle crossing analysis is carried out by establishing the mathematical model of front wheel obstacle crossing when the robot meets obstacles. To verify the analysis results, ADAMS software is used to simulate and analyze the process of robot running on the complex road surface and obstacles-crossing. Finally, a theoretical prototype is made to verify its feasibility. Theoretical analysis and experimental results show that the designed WLHR is feasible and has the stability of the wheeled mechanism and the higher obstacle crossing ability of the legged mechanism so that the robot can adapt to a variety of complex road conditions.

## 1. Introduction

Since the 1980s, mobile service robot technology has gradually become one of the research hotspots of researchers at home and abroad, and its application scope has also gradually expanded to space exploration [1], military reconnaissance, explosive disposal rescue [2], entertainment services, and other fields [3].

Nowadays, the ground mobile robots are the most widespread category of mobile device robots, and their application direction is far superior to that of industrial robots. For this research, a great amount of research has been done and a lot of effort has been made [4]. Ground mobile robots include wheeled robot, legged robot, and tracked robot. Wheeled robots can move simply and effectively at high speed and stably on flat and complex roads or sloping terrain [5], but in unstructured environments, the use of legged and tracked robots is also a valuable option. Tracked robots can move on rugged and uneven terrain, because their contact surface with the ground is much larger than wheeled and legged robots [6], which can make the operation more stable, but they usually move more slowly and with lower energy efficiency.

Among so many mobile mechanisms, wheeled and tracked robots are the most studied, but legged robots have also been researched because of their good obstacle crossing ability, such as BigDog [7]. However, because the leg structure is complex and highly integrated, its cost is also expensive, especially for the robot with dynamic gait planning, as the high complexity of its dynamic gait is related not only to control, but also to mechanical structure. Therefore, reducing the number of legs can simplify its complexity and reduce its cost [8]. At present, most legged robots are bipedal, quadruped, and hexapod [9]. However, due to the increasing requirements for application fields, for the common mobile robot it has been difficult to meet all requirements, so hybrid robots have been developed [10].

At present, the hybrid robot mainly has wheel-legged hybrid robot (WLHR), wheel-tracked hybrid robot (WTHR), wheel-track-leg hybrid robot (WTLHR), and so on. The WLHR has both the rapidity and smoothness of the wheeled robot and the superior obstacle performance of the legged robot. It can adjust its movement posture according to the external environment and is widely used in space exploration and military investigation [11]. According to the structure, it can be roughly categorized into wheel-leg connected type, wheel-leg separated type, and wheel-leg variant type [12]. Alper K [13], a transformable wheel-legged hybrid mobile robot, was proposed, which adopts a wheel-leg variant type. However, the structural design is complex and requires high control ability. Fei [14] proposed a WLHR, which can effectively improve the obstacle crossing ability of the robot through the clever combination of wheels and legs. The wheel-leg robot not only has the advantages of fast walking speed and flexible operation, but also the advantages of high obstacle crossing of the legged robot. Ben-Tzvi et al. proposed a WTHR [15], which improved the adaptability of the robot to the environment through the free switch of the two modes. The wheel-track-leg hybrid mobile robot [16,17] has good ground mobility and better obstacle-crossing ability. It is mostly suitable for obstacle-crossing in complex terrain, but its mechanical structure and control system is more complicated. Zhao [18] has studied a new type of WLHR, which combines the characteristics of quadruped mammals and realizes the wheeled, legged, and wheel-legged compound motion modes respectively. Kelly [19] studied the gait of the robot ATHLETE, such as advancing, climbing, and obstacle-crossing. The robotic laboratory of Swiss Federal Institute of Technology in Zurich university has developed a quadruped WLHR [20] called “ANYmal”. It is a wheel-leg connected type robot, and it can be found that the ANYmal robot has a variety of different gaits, such as crawling, jumping, and fast running.

According to the project requirements, it needs to be considered that the designed WLHR should have fast passing ability, certain obstacle crossing ability, low cost, and relatively simple control system. Considering the stability, three is the minimum number of legs for the mobile robot to maintain static stability, and considering the requirement for fast passing ability and simple control, we finally chose the quadruped leg structure combined with wheel, that is, the wheel-leg connected type hybrid robot. Although the above-mentioned mechanisms have significantly improved the obstacle crossing ability [21] and wide adaptability, they still have a large quality and complex control system. Therefore, we designed a low-cost, low-power, and modular WLHR to improve the robot’s mechanical robustness [22].

Based on the analysis of the mechanical structure and environmental adaptability of the WLHR, this paper proposes a relatively lightweight WLHR according to the required working environment of the robot. Its wheel-leg mechanism is based on a slider-crank mechanism. Compared with the wheel-leg mechanism mentioned in the above literature, such as [13,20], it has a simple design method, convenient control, and strong operability. The robot can transform the wheel and leg movement to adapt to the different road conditions: through the wheels to realize high-speed and long-distance movement and through the legs to cross obstacles and to adapt to complex terrain environments.

## 2. System Architecture and Design of Wheel-Legged Hybrid Robot (WLHR)

### 2.1. Conceptual Design of Robot

The WLHR designed in this paper is mainly composed of a mechanical system and a control system [23], as shown in Figure 1, which is a three-dimensional (3D) model of the mechanical structure of the robot.

The WLHR is equipped with four wheel-leg suspension devices symmetrically on the left and right sides of the car body, and each wheel has rubber tires with patterns on the outside to increase the friction between the wheels and the ground, and prevent the skidding phenomenon in the process of obstacle crossing. Each wheel-leg suspension device is equipped with a DC brushless motor, which is driven by C620 and placed on the inside of the car body. The ultrasonic sensor is placed on the front baffle and rear baffle of WLHR to sense the existence of obstacles. The linear actuator is placed on the front side of the WLHR and fixed with the two suspension devices in front of the robot to drive the movement of the leg suspension mechanism.

Considering the quality problem of the linear actuator, if the lithium cell is placed in the middle position, it may cause the center of gravity to move forward. To prevent the WLHR from overturning due to the unstable center of gravity in process of movement and obstacle crossing, the lithium cell should be placed on the rear side of the platform. Place the master control board in the center of the platform to facilitate connection with various electrical devices.

### 2.2. Suspension Device Design

The composition diagram of the suspension device is shown in Figure 2.

The suspension device is placed in the body, which is composed of a brushless DC motor, shock absorber rocker, cantilever bearing seat, main cantilever, and wheel bearing. The motor is connected with the cantilever bearing seat, and the motor is provided with a power input shaft, which is connected with one end of the transmission shaft through a coupling to drive the transmission shaft to rotate; the power take-off (PTO) shaft is connected with the bearing at the wheel to drive the wheel to rotate.

The leg transmission component, namely the main cantilever, is connected between the transmission shaft and the PTO shaft, and it mainly comprises a first connecting shell and a second connecting shell which are matched and connected, and an installation cavity which is defined by the first connecting shell and the second connecting shell; the driving wheel group, namely two synchronous wheels installed in the cavity, which drive the PTO shaft to rotate through the synchronous belt; and the first belt wheel, the second belt wheel, and the synchronous belt. The radial dimension of the first belt wheel is smaller than that of the second belt wheel; the synchronous belt is connected between the first belt wheel and the second belt wheel; the first belt wheel is connected with the transmission shaft; and the second belt wheel is connected with the PTO shaft.

## 3. Kinematics Analysis

### 3.1. Steering Analysis

The WLHR turns left and right through differential motion, and its motion state needs to calculate the speed of the left and right wheels.

To facilitate modeling, suppose:The car body and wheel-legs suspension device are rigid body, and the center of gravity is symmetrical;Each wheel has pure rolling with the ground without slipping or longitudinal sliding;Under complex road conditions, the front and rear wheels are controlled by the same and direction. By calculating the speeds of the front or back wheels, the rotational speed of all the wheels and the movement speed and direction of the robot body can be obtained.

Establish a kinematic model for the turning process of the mechanism, where $\left\{O_{T},X_{T},Y_{T}\right\}$ represents the fixed coordinate system of the mechanism. The letter symbols used in the analysis process are shown in Table 1.

The known formula of wheel speed is $V_{i}=\omega_{i}R_{i}$. When the speed of left and right wheels is the same, the robot speed is $V_{i}=V_{L}=V_{R}$; when the speed of the left and right wheels is not equal, the differential steering occurs, the WLHR makes differential steering motion.

As shown in Figure 3, when the left and right sides of the WLHR have the same speed direction but different sizes, the WLHR turns with a large radius, that is $R_{i}>D/2$.

Suppose the speed of the robot is $V_{i}$, that is
(1)$$V_{i}=\left(\frac{V_{L}+V_{R}}{2}\right)$$

It can be seen from the figure that the angular velocity of the WLHR around the steering center O point can be obtained from the triangle similarity, that is
(2)$$\omega_{i}=\frac{V_{i}}{R_{i}}=\frac{V_{R}}{R_{i}+D/2}=\frac{V_{L}}{R_{i}-D/2}=\frac{V_{R}-V_{L}}{D}$$

According to the above Formulas (1) and (2), the theoretical turning radius of the WLHR is
(3)$$R_{i}=\frac{D\left(V_{R}+V_{L}\right)}{2\left(V_{R}-V_{L}\right)}$$

As shown in Figure 4, when the speed of the left and right sides of the WLHR is opposite, the WLHR turns with a small radius, and the turning radius is less than the width of the car body, that is $0<R_{i}<D/2$.

At this time, the speed of the WLHR is
(4)$$V_{i}=\left(\frac{V_{L}-V_{R}}{2}\right)$$

The angular velocity is
(5)$$\omega_{i}=\frac{V_{i}}{R_{i}}=\frac{V_{R}}{R_{i}+D/2}=\frac{V_{L}}{D/2-R_{i}}=\frac{V_{R}+V_{L}}{D}$$

According to the above Formulas (4) and (5), the theoretical turning radius of the WLHR is
(6)$$R_{i}=\frac{D\left(V_{R}-V_{L}\right)}{2\left(V_{R}+V_{L}\right)}$$

### 3.2. Kinematic Model

As shown in Figure 5, the natural coordinate system {O,X,Y} is established in the plane to describe the pose equation at any time, and the local reference coordinate system $\left\{O_{T},X_{T},Y_{T}\right\}$ is established as the fixed coordinate system of the mechanism, taking the midpoint of the front left and right wheels as O, and the coordinates in the geodetic coordinate system as (X,Y).

From the knowledge of kinematics, the velocity can be obtained by deriving the displacement, and the angular velocity can be obtained by deriving the velocity.

The kinematics equation derived from the rotation matrix can be obtained as follows:(7)$$\left[\begin{array}{c} v_{x}\\ v_{y}\\ \omega \end{array}\right]=\left[\begin{array}{cc} cos\theta & 0\\ sin\theta & 0\\ 0 & 1 \end{array}\right]\left[\begin{array}{c} v_{i}\\ \omega_{i} \end{array}\right]$$

Bring Formulas (1) and (2) or Formulas (3) and (4) into Formula (7) to obtain:(8)$$\left[\begin{array}{c} v_{x}\\ v_{y}\\ \omega \end{array}\right]=\left[\begin{array}{cc} \frac{r}{2}cos\theta & \frac{r}{2}cos\theta \\ \frac{r}{2}sin\theta & \frac{r}{2}cos\theta \\ -\frac{R}{D} & \frac{R}{D} \end{array}\right]\left[\begin{array}{c} \omega_{L}\\ \omega_{R} \end{array}\right]$$

Taking time *t* as a variable, the pose equation of the legged robot at any time can be obtained:(9)$$\{\begin{array}{c} x\left(t\right)=x\left(0\right)+\frac{r}{2}\int_{0}^{t}\left(\omega_{L}+\omega_{R}\right)cos\theta \left(t\right)dt\\ y\left(t\right)=y\left(0\right)+\frac{r}{2}\int_{0}^{t}\left(\omega_{L}+\omega_{R}\right)sin\theta \left(t\right)dt\\ \theta \left(t\right)=\theta \left(0\right)+r\int_{0}^{t}\frac{\left(\omega_{L}+\omega_{R}\right)}{D}dt \end{array}$$

In the above formulas, (x(0),y(0),θ(0)) is the initial pose equation of WLHR.

According to the above kinematic analysis, the mechanism operation principle of the WLHR is simple, and its mobility and steering are excellent. The position of the robot at any time is related to the overall width of the mobile device and the rotation speed of the left and right wheels.

## 4. Simulation and Analysis

ADAMS software is used to simulate the kinematics of the WLHR. Firstly, the 3D model established in Solidworks is imported into Adams, and the multiple models are simplified without affecting the results, which reduces the operation difficulty, and the simulation results are analyzed accordingly. The main technical parameters of the mechanism are shown in Table 2.

### 4.1. Obstacle-Crossing Process Analysis

Climbing the stairs is a process in which the robot is continuously crossing obstacles, and it is also one of the difficult tasks that the all-terrain robot must complete [24,25]. In addition to the power of the robot itself, the key factors that influence the robot climbing the stairs are the torque and the specific parameters of the stairs. In the simulation model, the given stair height is 150 mm, and the stairs width is 260 mm. Referring to relevant design data, it can be seen that the designed stairs meet the ladder size of national standards. The process of the WLHR climbing the stairs is shown in Figure 6.

WLHR uses a waterproof ultrasonic sensor to sense the existence of obstacles, and its blind area has broken through to 13.8 cm, with good sensing performance. Through the analysis of the obstacle crossing process of the robot, it can be seen that when the robot encounters obstacles in the process of moving, the ultrasonic sensor senses the existence of the obstacle, so that the robot stops moving forward at a certain distance from the obstacle, and the front wheel is lifted under the action of the suspension device. Assuming that the right-front wheel is lifted, WLHR continues to move forward for a certain distance to make the right-front wheel contact with the obstacle and generate interaction force. At this time, the right-front wheel moves and swings the leg back to lift WLHR. At the same time, the left-front wheel quickly completes the actions of front swing leg and rear swing leg, and completes the action of obstacle crossing together with the right-front wheel. If the front wheel can cross the obstacle smoothly, it can drive the fuselage and other wheels to cross the obstacle; if the front wheel cannot cross the obstacle, the whole body will fail. Therefore, when the size of the whole robot is limited in a certain range, the obstacle-crossing performance of the front wheel-legs of the robot will directly affect the obstacle-crossing ability of the whole robot.

### 4.2. Analysis of Obstacle Crossing Height

To analyze the obstacle crossing performance, it is necessary to study the obstacle crossing height of the front wheel of WLHR. Because the slider-crank mechanism is used in the structural design of the wheel-leg suspension mechanism, it is easier to simplify it into a mathematical model to analyze the obstacle crossing height.

Shown in Figure 7 is the mathematical model analysis diagram of obstacle crossing height of front wheel legs.

The linear actuator is the slider A, the segment BC is the crank, and the segment AB is the connecting rod. The WLHR designed in this paper takes slider A as the driving links and crank BC as the driven links. Because its structure is a slider-crank mechanism, the existence of dead point position must be considered, and the forward stroke of linear actuator cannot exceed the dead point position. When the driven links crank BC is collinear with the connecting rod AB, the dead point position appears, that is, the segment $A_{1}B_{1}C$.

When WLHR is ready to cross the obstacle, whether the left-front wheel or the right-front wheel swings first, the position of the un-lifted wheel in the vertical direction cannot exceed the position of the lifted wheel in the vertical direction, otherwise the swinging wheel will not touch the obstacle.

It is assumed that the position below the front side of the wheel in contact with the obstacle is taken as the obstacle crossing height. It can be seen from Figure 7 that the height of the contact point E between the wheel and the obstacle represented by the green line is the maximum height that WLHR can actively cross the obstacle.

The letter symbols used in the analysis process are shown in Table 3.

According to the knowledge of trigonometric function:(10)$$H_{max}=R+h_{1}+h_{2}$$
(11)$$D_{max}=R+h_{1}+h_{3}$$
(12)$$h_{1}=sin\varepsilon \times L_{3}$$
(13)θ=ε
(14)$$h_{1}=sin\theta \times L_{3}$$
(15)$$h_{3}=sin\left(\beta -\alpha \right)\times L_{3}$$

According to the actual measured value, α=5°, β=65°, ε=5°, $L_{1}=190\;\mathrm{mm}$, $L_{2}=70\;\mathrm{mm}$, $L_{3}=120\;\mathrm{mm}$. Bringing the actual measured values into Equations (10)–(15), we can calculate that $H_{max}=281\;\mathrm{mm}$, $D_{max}=300\;\mathrm{mm}$. Since the wheel needs to have contact force with obstacles, the climbing height needs to be less than 281 mm.

It can be seen from the above analysis that the maximum obstacle crossing height of WLHR is related to factors such as the length of suspension device, wheel radius, and its related included angle.

### 4.3. Simulation Experiment of WLHR on the Complex Road Surface

Stability is an important index to evaluate the operation ability of WLHR [26]. Locomotion on flat and complex roads or slope terrain can be simply and effectively performed by wheels with high speed and energy efficiency. When crossing higher obstacles such as stairs, according to the process described in the obstacle crossing process diagram in Section 4.1, the obstacles can be stably crossed by mutual movement of wheels and legs.

Figure 8 shows the simulation process in ADAMS. Figure 8a provides the simulation diagram of the operation process of the robot on the complex road surface, including concave and convex slopes and obstacles. Figure 8b shows the simulation diagram of the operation process of the robot climbing stairs. Figure 8c shows the simulation diagram of the operation process of the robot on slope terrain. Figure 8d shows the simulation diagram of the left and right wheels of the robot continuously crossing obstacles. After the simulation runs, the velocity, centroid displacement, and acceleration curves of the robot can be obtained through the post-processing module of the software.

Figure 9a shows the change curve of centroid displacement when the robot moves on the complex road surface. It can be seen from Figure 9a that the displacement of the robot normally crossing the complex road is roughly consistent with the height of obstacles, with little fluctuation.

Figure 9b shows the change curve of centroid acceleration when the robot moves on the complex road surface. It can be seen from Figure 9b that the acceleration changes obviously when the robot climbs over obstacles, the forward speed decreases when climbing the obstacle, and the speed returns to normal after climbing the obstacle.

Figure 10a shows the centroid displacement curve of the robot climbing the steps. It can be seen from Figure 10a that the robot climbs the steps for 3.2 min 5 s to 10 s, the robot climbs the steps steadily, and there is no obvious carton when climbing the stairs. The fluctuation of the robot centroid is small, which proves that the robot climbs the stairs with good stability.

Figure 10b shows the acceleration of the robot centroid of mass in X, Y, and Z directions. It can be seen from Figure 10b that the acceleration of the robot remains at approximately 0 while climbing the steps, indicating that the robot is almost advancing at a stable speed; there is a small sudden change in acceleration at 6.5 s, which is the moment when the front leg of the robot just touches the stairs; there is a sudden change in the acceleration from 6 s to 7 s; and three directions occur at the same time. Therefore, it is inferred that the contact between the rear wheel of the robot and the step surface produces relative sliding.

The slope angle of climbing simulation experiment is 35° and the height is 0.56 m, which meets the technical parameters required by the project. Figure 11a shows the centroid displacement curve of the robot in the moving process of slope terrain. WLHR starts moving in slope terrain at 5 s. According to the smoothness of the curve, it has good stability and almost no fluctuation in the centroid. Figure 11b shows the change process of centroid acceleration curve of the robot during moving in slope terrain. From 0 s to 12 s, the centroid acceleration is approximately 0, which reflects the good stability and fast passing ability of the robot when moving in slope terrain. At 12 s, there is a sudden change in centroid acceleration, which is the reason why the robot meets obstacles on the plane after going downhill. The simulation results are completely consistent with the actual movement.

Figure 12a shows the change curve of centroid displacement when both sides of the robot are obstacle crossing continuously. During the simulation period of 5 s to 12 s, the change of centroid displacement is almost the same when the left side of the robot continuously obstacle crossing and the right side continuously obstacle crossing, which verifies the rationality of WLHR symmetry design. It can be seen from Figure 12b that the centroid acceleration is also approximately equal to 0, and only slightly changes when meeting obstacles and obstacle crossing, which proves that WLHR has good lateral stability. Figure 12c shows the force changes of the left and right wheels during continuous obstacle crossing. When the left wheel continues to obstacle crossing, the force on the right wheel increases, and the force on the left wheel decreases. When the right wheel continues to obstacle crossing, the force on the left wheel increases, and the force on the right wheel decreases. This motion process conforms to the model that both sides of the robot continuously obstacle crossing, which verifies the stability of the robot.

The result proves that the robot can stably complete the motion on flat and complex roads, slope terrain, and the climbing steps and obstacle crossing functions, which verifies that the robot structure design is rational and simple.

Through simulation and analysis, compared with other WLHR proposed in the previous scientific literature, it has similarities and differences, but it still has its unique features.

The WLHR system designed in this paper has similarities with BIT-NAZA [10], such as independent driving wheel-leg mechanism and high stability but shows the following differences: (i) BIT-NAZA has a higher body height and better passing ability and (ii) both are symmetrical structures, but BIT-NAZA can change the symmetrical width of the legs.

Regarding the wheeled locomotion, there are some similarities with the ANYmal [20]: (i) The wheels are placed at the end of rotating arms and the steering motion adopts differential rotation; (ii) the most compact and robust design.

## 5. Prototype Verification

A prototype is made to verify the function and mobile ability of the proposed WLHR, as shown in Figure 13. Through the obstacle avoidance function test, the perception ability of the robot is verified. The mobile ability of the prototype was tested in different situations, including the vertical obstacle crossing experiment, continuous step climbing experiment, and grassland passing experiment.

### 5.1. Analysis of Obstacle Avoidance Experiment

This paper adopts an ultrasonic sensor, which can not only provide perception when obstacle crossing, but also provide obstacle avoidance function when WLHR cannot cross the obstacle.

The principle of obstacle avoidance of ultrasonic sensor is based on the relationship between sound velocity, time, and obstacle distance. The expressions of the three can be expressed as follows [27]:(16)$$S=\frac{CT}{2}$$

In the above expression, *C* represents the velocity of sound; *S* represents the distance between WLHR and obstacles; and *T* represents the time difference between the time of transmitting pulse and the time when the first echo arrives.

With the help of fuzzy reasoning and control, the accuracy of obstacle avoidance algorithm in application is strengthened, which is convenient for WLHR to identify and control obstacles. Taking the designed ultrasonic obstacle avoidance algorithm as an example, the main operation process of WLHR is shown in Figure 14.

As shown in Figure 14, when WLHR works, firstly, initialize the parameters and adjust the obstacle avoidance parameters of WLHR, so that the WLHR can select the obstacle free route within a safe distance in time. Then, the ultrasonic sensor is used to collect the acoustic signal around WLHR, detect the distance between WLHR and obstacles, compare the data with the set data, and select a reliable operation route. When meeting obstacles, WLHR avoids obstacles, rotates at an appropriate angle, and continues to choose safe driving. If WLHR does not detect obstacles, it can directly choose to go straight without obstacles to complete the specified tasks.

Figure 15 shows the sensing diagram of WLHR ultrasonic sensor.

The beam angle of the ultrasonic sensor in the vertical direction is 60°, and the installation position is at the front baffle of WLHR, with a height of 280 mm from the ground. The height between the car body and the ground designed in this paper is 300 mm. The maximum obstacle crossing height is $H_{max}=281\;\mathrm{mm}$, which is just the same as the installation height of ultrasonic sensor. Therefore, when the sensing height of the ultrasonic sensor is higher than 280 mm, WLHR avoids obstacles; when the sensing height of the ultrasonic sensor is lower than 280 mm, WLHR will cross the obstacle.

Sensing the distance of obstacles is the key to obstacle avoidance. Therefore, it is very important to test the obstacle avoidance distance of the prototype. Shown in Table 4 is a test of the distance of perceptible obstacles.

It can be seen from the table that WLHR has high perception sensitivity, and the maximum distance of obstacles that can be detected is 400 cm.

Since WLHR is differential motion, it is necessary to consider the appropriate position to stop when sensing obstacles. Otherwise, WLHR will collide with the obstacle during turning.

In the prototype obstacle avoidance experiment, it is assumed that the width of the obstacle is the same as that of WLHR, which is D. The test environment of the prototype is set on a relatively smooth ground. Because the friction coefficient varies according to different terrain, the minimum turning distance during obstacle avoidance will also be different.

Shown in Figure 16 is the minimum distance experiment that the prototype can turn successfully when avoiding obstacles. Shown in Table 5 is the minimum obstacle avoidance distance in which WLHR can complete turning under the condition of width D.

### 5.2. Prototype Experiment in Complex Terrain Environment

As shown in Figure 17, the robot is conducting a vertical obstacle crossing experiment, and the linear actuator pushes the suspension device to drive the wheel to lift through the obstacle. The experiment shows that the obstacle crossing height of the mechanism is 150 mm.

As shown in Figure 18, the robot is conducting the continuous step climbing experiment. The experiment shows that the WLHR has the ability of continuous obstacle crossing, and the dimension of continuous step is 250 mm tread wide and 160 mm tread heigh.

As shown in Figure 19, the robot passes the experiment in the grassland. The elastic restoring force of the self-adaptive mechanism can make the WLHR walk stably in the irregular terrain, and the fluctuation of the centroid is small, which proves that the mechanism has good passing ability in the complex terrain.

Through the prototype experiment, it is found that the WLHR has excellent adaptive ability and sensitive perception ability, and it has high trafficability on the various complex ground, which meets the all-terrain operation conditions. The motion process is consistent with the theoretical analysis.

## 6. Conclusions

In this paper, a WLHR that can adapt to an all-terrain environment was designed and implemented. It has both the mobility of wheel robot and the obstacle crossing ability of leg robot. The research process was as follows:

Firstly, the overall structure design and steering performance analysis were carried out, and the kinematics model was established to analyze the motion state of the robot.

Secondly, the maximum height of WLHR obstacle crossing was analyzed by mathematical modeling, and the maximum height was obtained. The kinematics simulation of the WLHR in different environments was carried out by ADAMS software, and the corresponding kinematics curves were obtained.

Finally, the experiments of obstacle avoidance and fast walking on flat ground, steering, obstacle crossing, and stair climbing were carried out on the prototype robot, which verifies the feasibility and practicability of the WLHR designed in this paper.

## Figures and Tables

**Figure 1 sensors-21-05367-f001:**
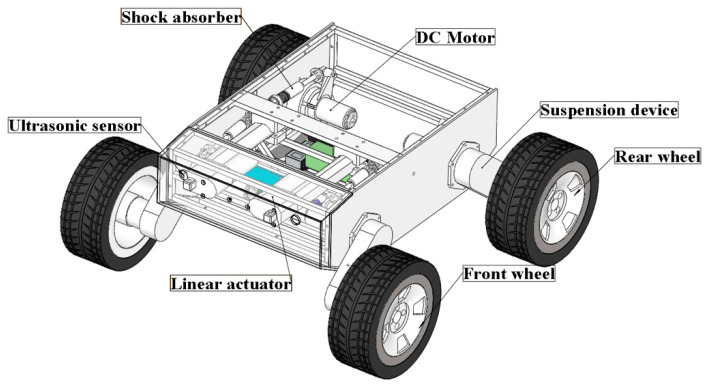
Structure of deformable wheel-legged hybrid robot (WLHR).

**Figure 2 sensors-21-05367-f002:**
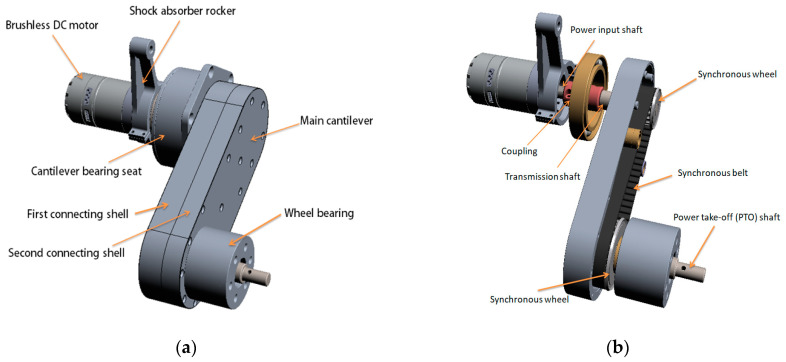
3D diagram of suspension device: (**a**) external structure; (**b**) internal structure.

**Figure 3 sensors-21-05367-f003:**
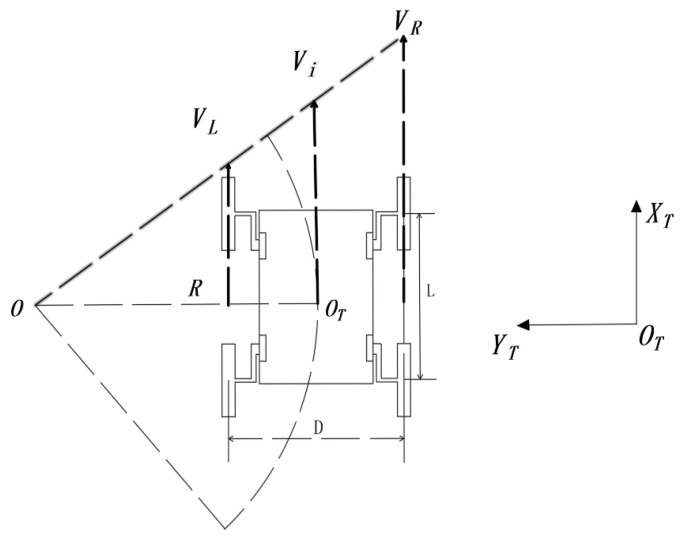
Large radius turning process.

**Figure 4 sensors-21-05367-f004:**
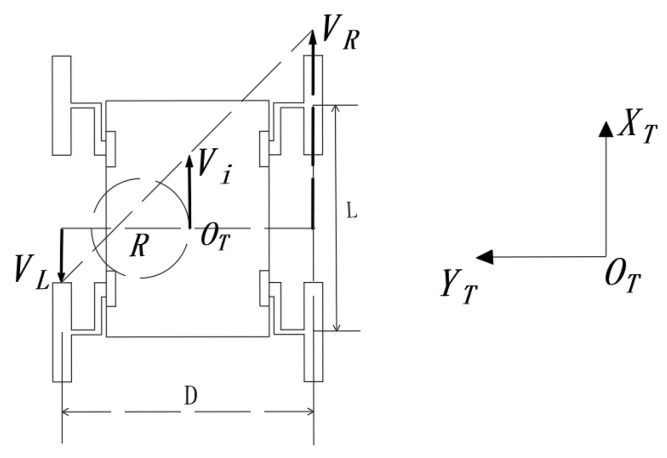
Small radius turning process.

**Figure 5 sensors-21-05367-f005:**
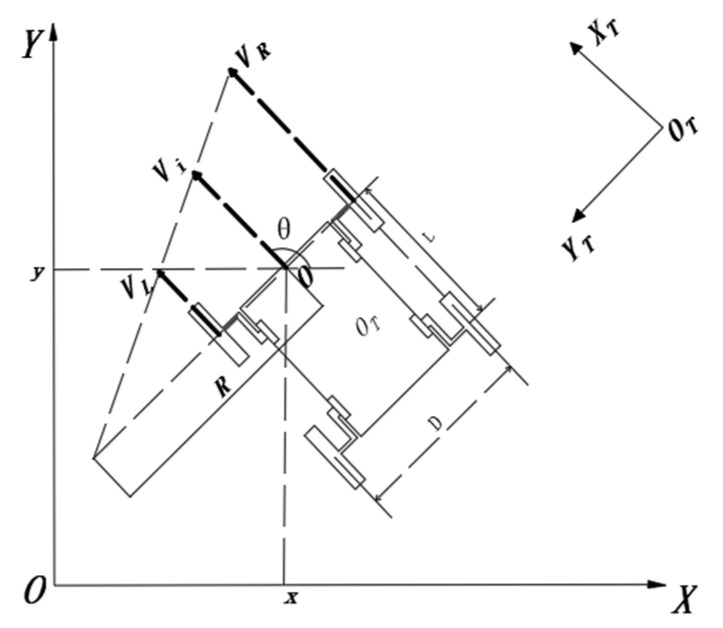
Kinematics model.

**Figure 6 sensors-21-05367-f006:**
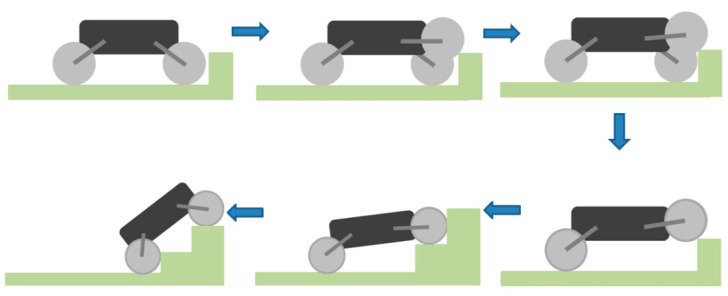
Diagram of obstacle crossing process of WLHR.

**Figure 7 sensors-21-05367-f007:**
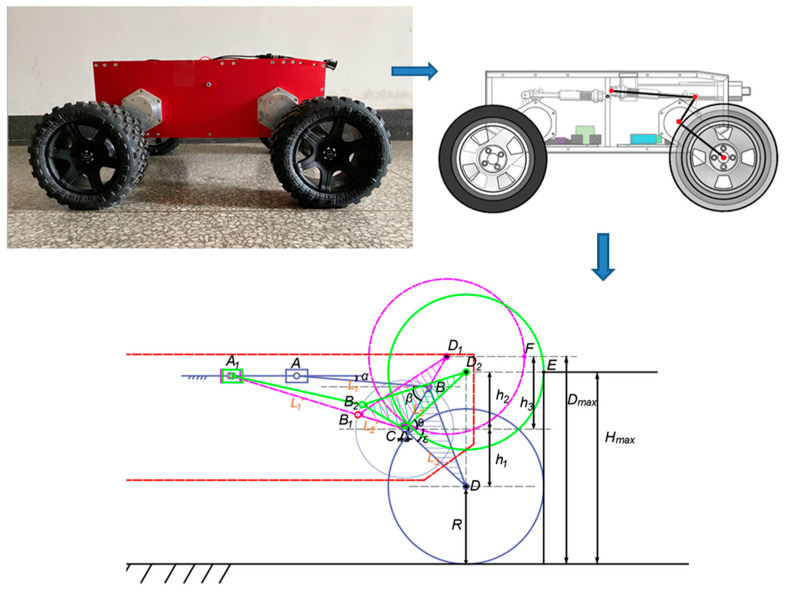
Mathematical model analysis of obstacle crossing height.

**Figure 8 sensors-21-05367-f008:**
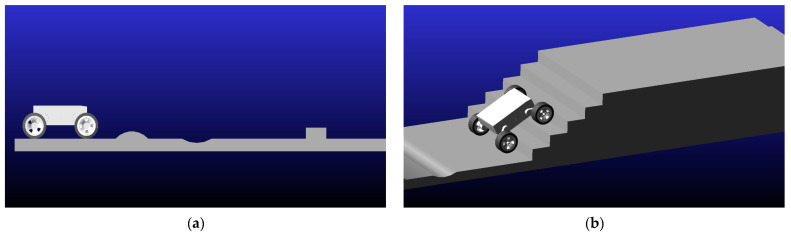
ADAMS simulation diagram: (**a**) simulation drawing of crossing complex road surface; (**b**) simulation drawing of climbing stairs; (**c**) simulation drawing of moving in sloped terrain; (**d**) simulation drawing of both sides continuous obstacle crossing.

**Figure 9 sensors-21-05367-f009:**
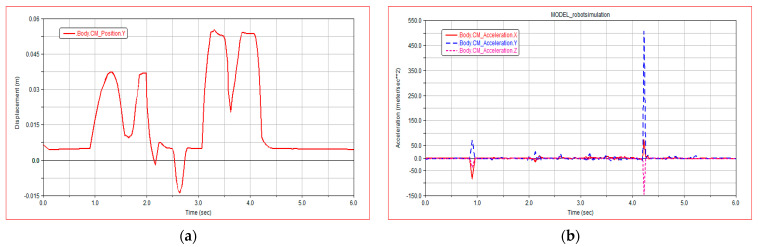
Simulation result diagram of crossing complex road surface: (**a**) robot centroid displacement; (**b**) robot centroid acceleration.

**Figure 10 sensors-21-05367-f010:**
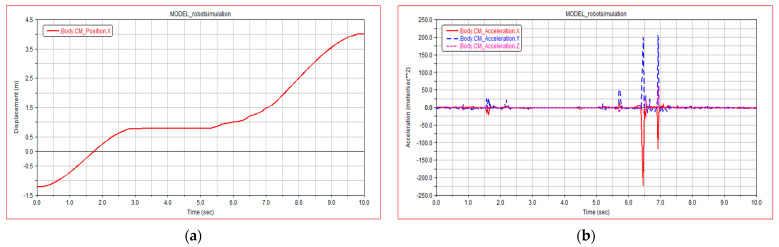
Simulation results of climbing stairs: (**a**) robot centroid displacement; (**b**) robot centroid acceleration.

**Figure 11 sensors-21-05367-f011:**
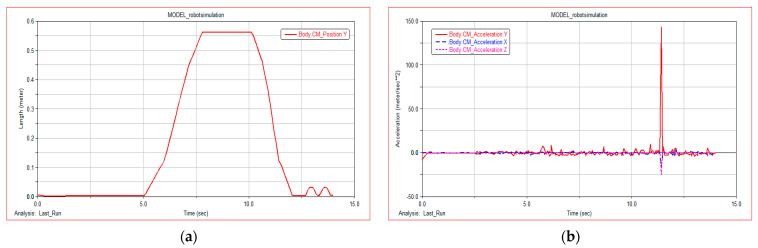
Simulation result of moving in slope terrain: (**a**) robot centroid displacement; (**b**) robot centroid acceleration.

**Figure 12 sensors-21-05367-f012:**
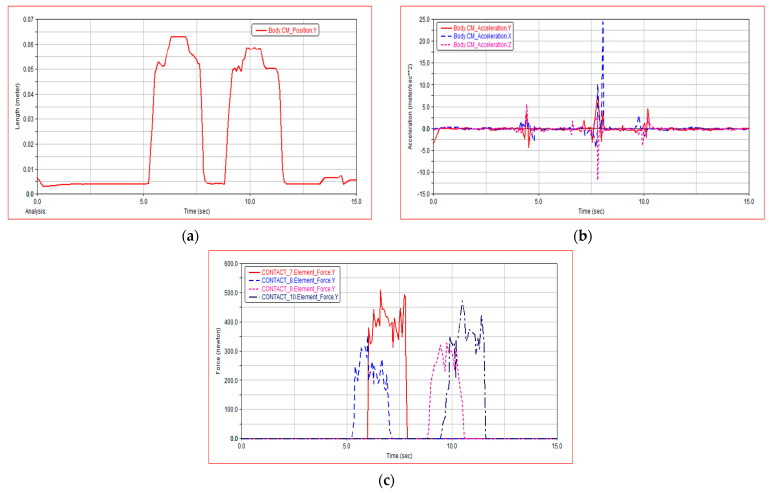
Simulation result of both sides continuously obstacle crossing: (**a**) robot centroid displacement; (**b**) robot centroid acceleration; (**c**) force on the ground contact point of left and right wheels in Y direction.

**Figure 13 sensors-21-05367-f013:**
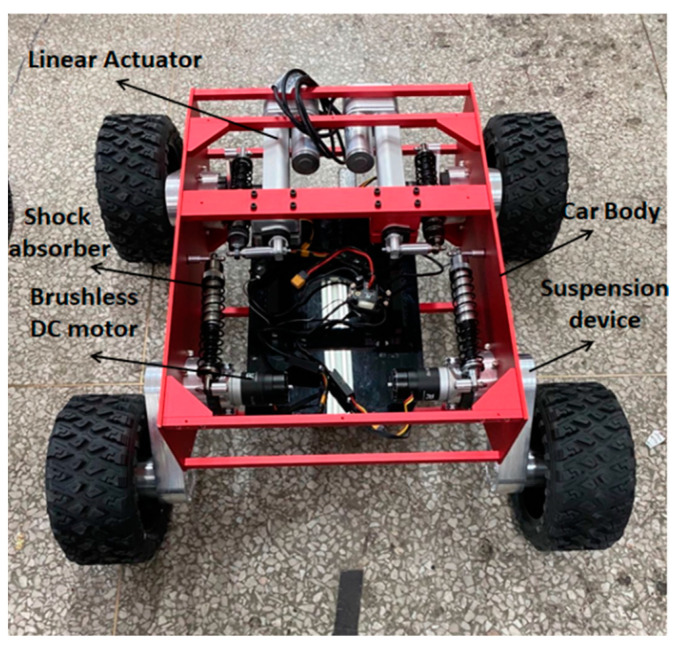
Prototype of the WLHR.

**Figure 14 sensors-21-05367-f014:**
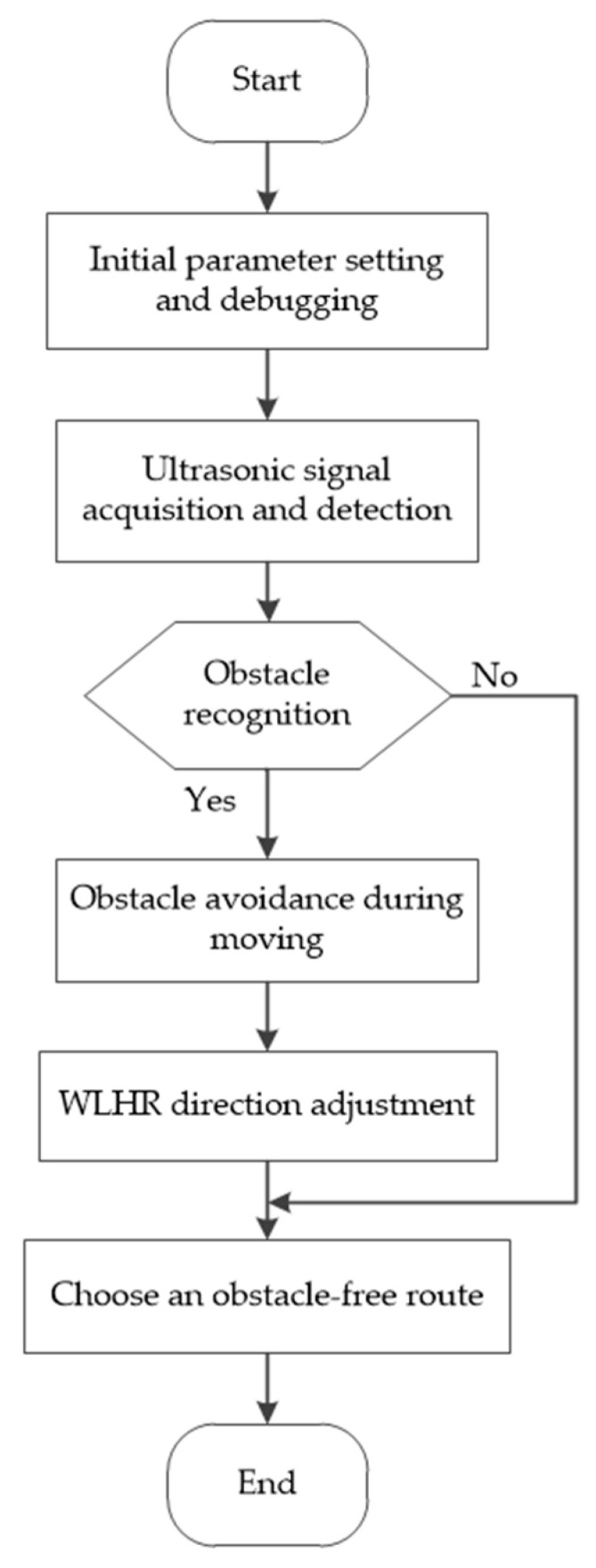
Operation flow chart of obstacle avoidance algorithm.

**Figure 15 sensors-21-05367-f015:**
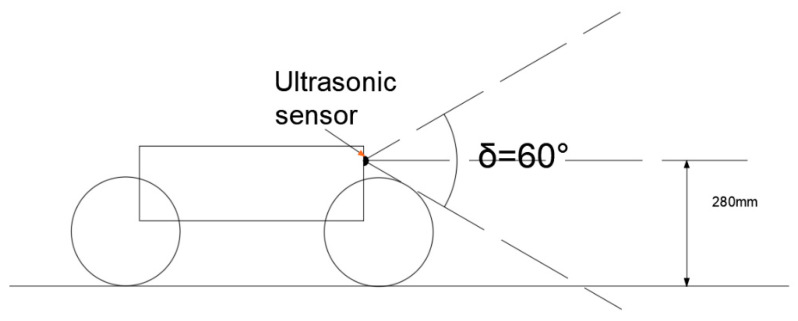
Schematic diagram of ultrasonic sensor sensing.

**Figure 16 sensors-21-05367-f016:**
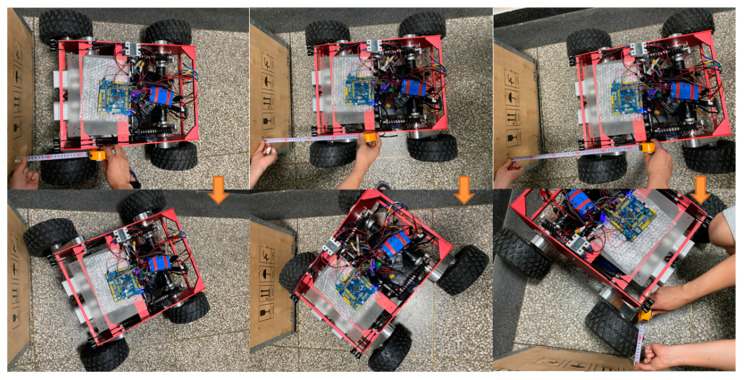
Obstacle avoidance and turning experiment of prototype.

**Figure 17 sensors-21-05367-f017:**
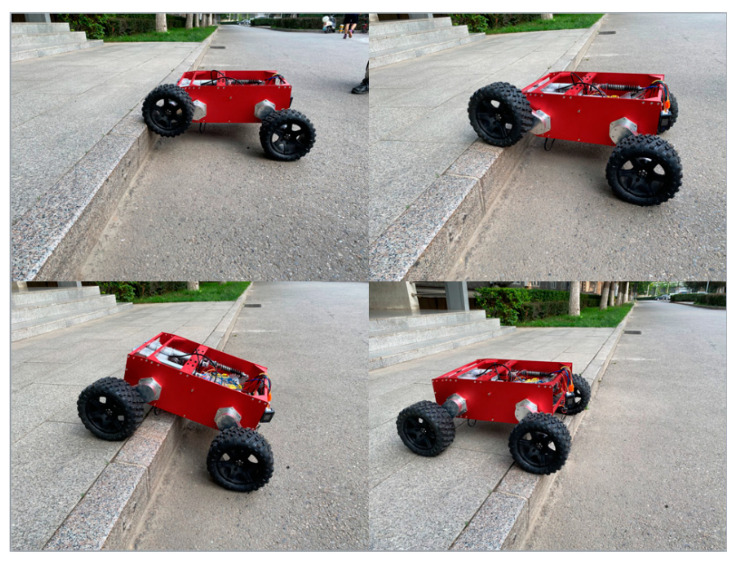
Robot vertical obstacle crossing experiment.

**Figure 18 sensors-21-05367-f018:**
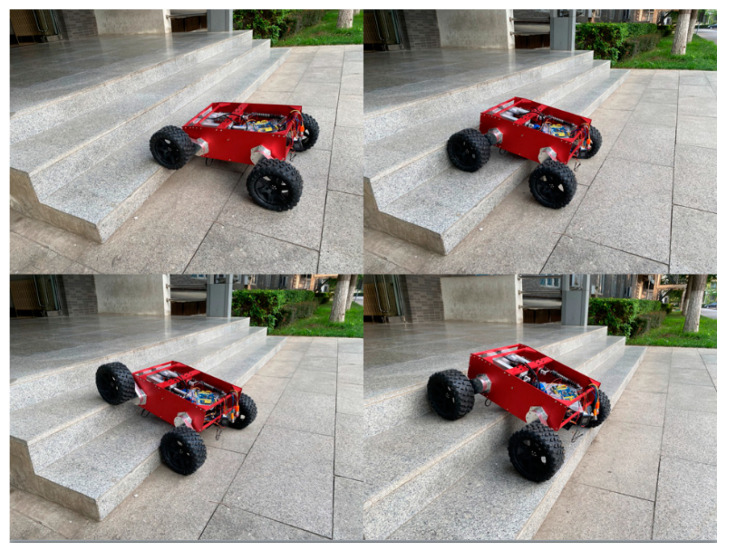
Robot vertical obstacle crossing experiment.

**Figure 19 sensors-21-05367-f019:**
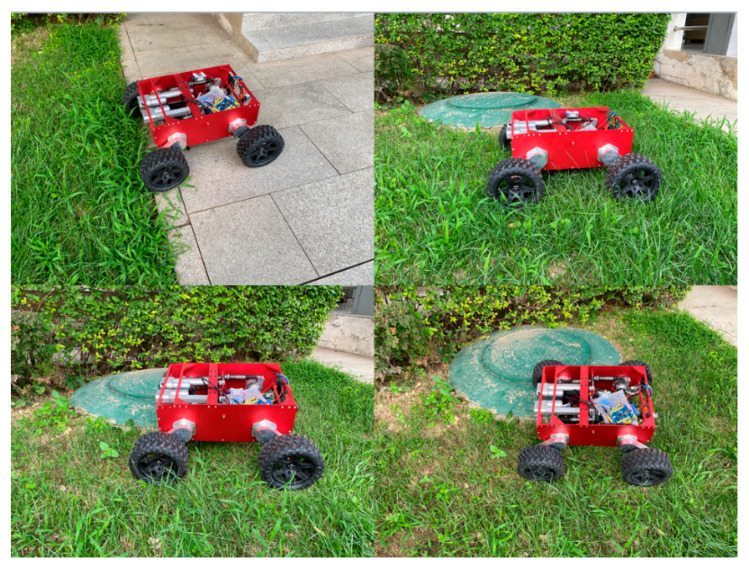
Robot grassland passed the experiment.

**Table 1 sensors-21-05367-t001:** Symbol list.

Symbol	Meaning
*R*	Theoretical turning radius
*D*	Left and right wheel track
$\omega_{i}$	Instantaneous angular velocity at $O_{T}$
$\omega_{R}$	Right wheel angular velocity
$\omega_{L}$	Left wheel angular velocity
$V_{i}$	Velocity of point $O_{T}$ relative to the ground
$V_{L}$	Outside linear velocity of left wheel
$V_{R}$	Outside linear velocity of right wheel
$O_{T}$	Centroid position
O	Center line of front left and right wheels
θ	Angle between body and ground

**Table 2 sensors-21-05367-t002:** Main technical parameters of the mechanism.

Technical Index	Parameter
Structure size of mechanism (mm)	Theoretical turning radius
Size of suspension mechanism (mm)	190 × 70 × 35
Wheel radius (mm)	111
Total mass (kg)	20~35
Platform load (kg)	≤20
Motor speed(rpm)	469
Motor output torque (N · mm)	3000
Battery life (h)	4
Climbing ability	≥30°
Vertical height of obstacle crossing (mm)	≤200
Supply voltage (V)	24
Shatter resistance	Falling at a height of 3 m, the structure is not damaged
Control mode	Remote control/autonomous control
Terrain environment	Flat ground, vertical obstacle, grassland, mine road, etc.

**Table 3 sensors-21-05367-t003:** Symbol list.

Symbol	Meaning
*R*	Wheel radius
$L_{1}$	Length of shock absorber
$L_{2}$	Length of shock absorber rocker
$L_{3}$	Length of connecting shall
α	The angle between linear actuator and shock absorber
β	The angle between shock absorber rocker and shock absorber
ε	The angle between suspension device and the horizontal position the car body
$D_{max}$	The maximum height that the wheel can lift when the dead point position appears
$H_{max}$	Maximum height of obstacle crossing
D	Wheel midpoint

**Table 4 sensors-21-05367-t004:** Distance of perceptible obstacles.

Distance (cm)	Result (Yes/No)
15	Yes
25	Yes
40	Yes
60	Yes
80	Yes
100	Yes
150	Yes
200	Yes
300	Yes
400	Yes
>400	No

**Table 5 sensors-21-05367-t005:** Complete the minimum distance test of obstacle avoidance function.

Obstacle Width (mm)	Minimum Distance (mm)	Success Rate (%)
D	100	0
	150	0
	200	0
	250	0
	300	100
	>300	100

## Data Availability

The data are available upon request.

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
