# Peer review of "Design and Research of All-Terrain Wheel-Legged Robot"

_sensors, 2021, doi:10.3390/s21165367_

Round 1

Reviewer 1 Report

In this paper, an all-terrain wheel-legged hybrid robot (WLHR) is developed, which can realize the transformation of wheel movement and leg movement to improve the adaptability to road environment. The structure of the WLHR is designed, the kinematics mode of the WLHR is established, and the obstacles-crossing ability of the WLHR is analyzed through ADAMS simulation. Additionally, a WLHR prototype is developed, and the mobile performance of the prototype is tested in different road situations. The ADAMS simulation and experimental results show that the WLHR has higher obstacle crossing ability, and hence adapts to complex road conditions.

The following suggestions are listed for the authors’ reference to further improve the paper:

  1. Check the structure of the suspension devices of rear wheels shown in Figure 1, they are different from the ones in other Figures.
  2. If the words “mobile mechanism” and “wheel-legged hybrid robot” have the same meaning, the “wheel-legged hybrid robot” is suggested in this paper.
  3. “Prototype of the WLHR” is suggested and used as the title of Figure 10.
  4. Check the styles of variable symbols in equations (1) and (4), they are different from the ones in other equations.

Author Response

Thanks to the editor for arranging you to review the manuscript for me, and thank you for your valuable comments. I have carefully answered the questions one by one in accordance with your suggestions and requirements, and carefully revised the paper.

Reviewer 2 Report

1. The authors proposed a mechanism design of WLHR robot which adapts to all-terrain environment. Each wheel of the robot is attached to a leg with 1 dof. Kinematic model of the robot was derived. Obstacle-crossing ability of the proposed robot was tested using dynamic simulation. A working prototype of the WLHR was constructed to evaluate its obstacle crossing ability under various situations.

2. The novelty of the WLHR design is not highlighted compared to other wheel-leg, wheel-track, wheel-leg-track hybrid design. In fact, the proposed design is simple and its merits are not explained and demonstrated clearly in simulation and experiments compared to other WLHR design.

3. The manuveurability and obstacle crossing ability of the proposed design was not compared with those of other WLHR design reported in literature. It is hard to judge how well the proposed design has performed compared to existing design. The authors are suggested they compare the performance of various WLHR design using simulation to illustrate the superiority of the proposed design.

4. How are obstacles sensed in the robot?

5. What is the maximum obstacle height that the robot can cross over? What are the factors (e.g length of cantilever of leg) of the design that affect the maximum ostacle height that the robot can cross? I suppose this can be modelled mathematically.

6. Basically, there is no quantitative analysis of the performance of the working prototype included in the manuscript.

7. Is it possible to extend the design to maintain the tilt angle of the robot body during obstacle crossing?

Author Response

(The authors gave the same response as above.)

Round 2

Reviewer 2 Report

The authors have addressed most of my comments from the previous review. However, the authors are suggested that they elaborate on how the ultrasonic sensor data is exploited in obstacle crossing control in terms of control algorithm or flow chart.

Author Response

Thank you again for your valuable comments. I have carefully answered the questions  in accordance with your suggestions and requirements, and carefully revised the paper.

Response 1: Thank you for your suggestion. According to your suggestion, in Chapter 5.1, I added the analysis of how the ultrasonic sensor obstacle crossing, such as ultrasonic principle and operation process, and added the flow chart.